# Innovative Phospholipid Carriers: A Viable Strategy to Counteract Antimicrobial Resistance

**DOI:** 10.3390/ijms242115934

**Published:** 2023-11-03

**Authors:** Daria Nicolosi, Giulio Petronio Petronio, Stefano Russo, Maria Di Naro, Marco Alfio Cutuli, Claudio Russo, Roberto Di Marco

**Affiliations:** 1Department of Drug and Health Sciences, Università degli Studi di Catania, 95125 Catania, Italy; dnicolosi@unict.it (D.N.); maria.dinaro14@gmail.com (M.D.N.); 2Department of Medicine and Health Sciences “V. Tiberio”, Università degli Studi del Molise, 86100 Campobasso, Italy; giulio.petroniopetronio@unimol.it (G.P.P.); m.cutuli@studenti.unimol.it (M.A.C.); claudio.russo@unimol.it (C.R.); roberto.dimarco@unimol.it (R.D.M.); 3Division of Biochemistry, Medical Faculty Mannheim, Mannheim Institute for Innate Immunoscience (MI3), Heidelberg University (HBIGS), 68167 Mannheim, Germany; 4Consorzio Interuniversitario in Ingegneria e Medicina (COIIM), Azienda Sanitaria Regionale del Molise ASReM, UOC Governance del Farmaco, 86100 Campobasso, Italy

**Keywords:** antimicrobial resistance, antibiotics, phospholipids

## Abstract

The overuse and misuse of antibiotics have led to the emergence and spread of multidrug-resistant (MDR), extensively drug-resistant (XDR), and pan-drug-resistant (PDR) bacteria strains, usually associated with poorer patient outcomes and higher costs. In order to preserve the usefulness of these life-saving drugs, it is crucial to use them appropriately, as also recommended by the WHO. Moreover, innovative, safe, and more effective approaches are being investigated, aiming to revise drug treatments to improve their pharmacokinetics and distribution and to reduce the onset of drug resistance. Globally, to reduce the burden of antimicrobial resistance (AMR), guidelines and indications have been developed over time, aimed at narrowing the use and diminishing the environmental spread of these life-saving molecules by optimizing prescriptions, dosage, and times of use, as well as investing resources into obtaining innovative formulations with better pharmacokinetics, pharmacodynamics, and therapeutic results. This has led to the development of new nano-formulations as drug delivery vehicles, characterized by unique structural properties, biocompatible natures, and targeted activities such as state-of-the-art phospholipid particles generally grouped as liposomes, virosomes, and functionalized exosomes, which represent an attractive and innovative delivery approach. Liposomes and virosomes are chemically synthesized carriers that utilize phospholipids whose nature is predetermined based on their use, with a long track record as drug delivery systems. Exosomes are vesicles naturally released by cells, which utilize the lipids present in their cellular membranes only, and therefore, are highly biocompatible, with investigations as a delivery system having a more recent origin. This review will summarize the state of the art on microvesicle research, liposomes, virosomes, and exosomes, as useful and effective tools to tackle the threat of antibiotic resistance.

## 1. Introduction

Antibiotics are the staple of treatments against bacterial infections: since their discovery in the early 20th century, the death rates from threatening microbes lowered drastically. Antibiotics’ therapeutic effects involve either inhibiting proliferation (bacteriostatic) or killing pathogens (bactericidal) with the lowest affinity for the host’s cells; the mechanisms thanks to which antibiotics achieve their effect are diverse, extending from a disruption in the bacterial wall to an impairment in vital internal processes such as DNA decoding, RNA translation, and protein synthesis. Bacteria, however, can develop antimicrobial resistance (AMR) to antibiotics via simple natural selection: when they stay in enough contact with a chemotherapeutic drug, selected strains can produce genes that grant themselves endurance against it; this event occurs in a relatively quick timespan, and sadly, the survival of multi-resistant strains has been fostered in recent years by the massive and more than often erroneous employment of antibiotics and by the worsening of the hygienic criteria in hospitals due to monetary cuts to healthcare budgets [1,2,3].

Nowadays, multidrug-resistant (MDR) bacteria represent one of the biggest threats to human health because of their lethality and innate capacity to pass over resistance-inducing genes [4]; even during the COVID-19 pandemic, MDR bacteria were detected as a co-infection factor in a large portion of hospitalized COVID patients and resulted in being one of the major contributing causes of worse outcomes and, eventually, death [4,5,6]. To prevent the chemo-resistance phenomenon and, consequentially, the rise of MDR bacteria, innovative phospholipid carriers (IPCs) were demonstrated to be promising in the improvement in the efficacy profile of antibiotics.

IPCs are third-generation, phospholipid-based pharmaceutical formulations that act as carriers for the delivery of a pharmacologically active substance. Their development aims at achieving both spatial and temporal modulation of the release of the active compound, control of the quantity of the substance to be encapsulated, and protection of the drug from metabolization and degradation reactions [7]. Among them, liposomes, functionalized or not; exosomes; and virosomes found broad application in the fight against chemo-resistance in both experimental and clinical settings: their principal advantage against traditional methods to overcome bacterial resistance resides in their capacity to make old, ineffective molecules efficacious again versus MDR bacteria, thus reducing the need for the synthesis of new antibiotics.

This review will evaluate the most recent advances against chemo-resistance by incorporating antibiotics or other molecules in IPCs.

### 1.1. Antibiotics and Chemo-Resistance: A Worrying Phenomenon

Antibiotics were formerly discovered by Sir Alexander Fleming in 1928, who noticed the antibacterial activity of a certain mold, *Penicillium notatum*, on different pathogenic microbes; his studies on molds were inspired by his precedent research on lysozyme and by the findings of medical captain Vincenzo Tiberio on the prominent efficacy of *Penicillium glaucum* against a wide range of bacteria [8].

In 1943, Fleming won the Nobel Prize for medicine. Only two years later, he started giving the world an important message that remained unheard: large-scale use, especially when erroneous, of penicillin and antibiotics, in general, could lead to the end of the miracle since it would accentuate the selection of resistant bacteria [9].

In the recent past, the situation escalated to unthinkable levels: chemo-resistance became one of the greatest concerns of the 21st century, to the point that some microbial species are completely immune to last-line antibiotics like vancomycin, thus re-acquiring an unprecedented lethality [10].

Antibiotics suppress microbial infection through a wide variety of bacteriostatic and bactericidal mechanisms (Figure 1 and Table 1): e.g., beta-lactams, like penicillins and cephalosporins, bind irreversibly to D-alanyl-D-alanine-transpeptidase (DD-TPs) and other carboxypeptidases (commonly known as penicillin-binding proteins, PBPs), inhibiting the formation of cross-links and so causing the disruption the bacterial wall, which leaves the target vulnerable to osmotic and molecular pressures [11].

AC stems from microbial selection because of the large-scale use of antibiotics but, most importantly, for their clinical misusage (which permits the bacteria to stay in contact with a non-lethal dose of antibiotic for enough time to develop resistance) [44] and environmental misusage (farms that feed the animals huge amounts of antibiotics make the selection progress faster both in animals and in consumers of the final product) [45]. Moreover, most antibiotics are excreted unmodified from animals or from industrial waste, and both byproducts are sadly poured into rivers, lakes, and terrains, which become gyms for bacteria to train in. MDR bacteria can be consistently found in nature [46,47]. Add the hygiene- and protocol-related problems in hospitals to make up the perfect storm, with the rise of multi-resistant nosocomial infections that nowadays spread even out of hospitals to become serious community infections [48,49].

Even if the awareness of the chemo-resistance problem is low, scientists looked for solutions against it. The first logical approach was to search for more natural and synthesized small molecules that retained their activity against resistant microbes. This strategy, although functional, proved in recent years to be destined for failure since even the newest drugs will eventually encounter resistance due to the bacterial high capacity in resistance gene development and transmission. The risk of untreatable microbial infections is so impervious that it requires a multidisciplinary, coordinated action of the scientific community and the final users, doctors, patients, farmers, or industries.

Antimicrobial peptide-based therapy, suggested by and developed from the multitude of insects, plants, and other organisms that make daily use of them for survival [50,51,52,53], showed a minor insurgence of antibiotic resistance due to the intrinsic multi-modal, multi-target action derived from the more complex structure of these compounds, and a plethora of co-adjuvant properties such as antioxidant, anti-inflammatory, and pro-healing effects [54,55]. However, their application in therapy is restricted because of their side effects, caused by off-target reactions with human epitopes and by the high concentration (derived from their conspicuous molecular weight) at which they need to be administered to exert their properties.

Another molecule-based approach, often involving peptides, is to recognize and thus inhibit the principal factors that trigger antibiotic resistance: notable examples are the elimination of biofilms, blockage of multidrug resistance pumps (which could be over-expressed both for genetic reasons or due to heavy metal contamination), amelioration of bacterial membrane permeability, and disruption of vital bacterial processes like dihydrofolate reductase (DHFR) and translational elongation of protein inhibition [56,57,58]. While useful, this solution alone in the long term is insufficient since it faces the same obstacles as the principal resistance mechanism, namely target mutation and escaping, thanks to the innate adaptability of the bacteria.

Recent renewed interest arose in the employment of metals, despite non-specificity, against resistant bacteria: not only have they been demonstrated to be exceptional carriers for traditional and innovative antibiotics since they possess intrinsic antibacterial properties, targeting capacity, and low immune response, but even metals alone as such, in particular metal ions, could be of synergistic usage to combat microbes [59,60,61].

From the engineering and agro-industrial point of view, removing pharmaceutics and antibiotic resistance genes (ARGs) from water, soil, animal feed, and manure would greatly help reduce antibiotic resistance. This feature can be achieved by combining metal-based filtering systems [62,63], biological and biotechnological agents [64,65], and better maintenance and management [66,67]; a remarkable, somewhat different example in the field is brought up by the use of cultivated meat, which not only drastically reduces CO_2_, biological, and chemical waste with respect to farms but also almost deletes the need for antibiotics [68,69].

Last but not least, awareness campaigns [70,71], surveys [72], and technological integration in work [73,74] and learning [75,76] environments are crucial to stop an otherwise announced disastrous, unstoppable plague: communication is the key to unified, efficacious actions, especially when the enemy to defeat is so protean, diffused, and challenging.

### 1.2. Innovative Phospholipid Carriers (IPCs)

In recent years, nanomedicine significantly ameliorated therapy against a wide variety of illnesses, thanks to the capacity of these innovative drug delivery systems to obtain a time-controlled, precise, and quantitatively major transport of actives to the pharmacological targets: such a heavy impact on pharmacokinetics permitted the resurrection of old molecules that became inactive for years or that were not even considered for clinical usage because of their high toxicity, and even alone, nanoparticles contribute to fostering innovation in the fields they are applied into [77,78,79].

Lipid nanoparticles are one of the most promising vehicles of the family thanks to their elevated degree of biocompatibility, encapsulation efficiency, and customization [80]. Among them, the two types frequently used for the treatment of microbial infections are solid lipid nanoparticles and liposomes [81,82]; this review will discuss the main features and some of their technological evolutions, namely functionalized liposomes and exosomes, under the label of innovative phospholipid carriers (IPCs).

#### 1.2.1. Liposomes

Liposomes are vesicles composed of phospholipids, which are naturally formed when the latter are inserted in aqueous solutions. They can be used both as biomembrane models to preliminarily assess the interaction of compounds in terms of quantity and quality and as drug carriers [83,84].

Phospholipids are amphiphilic compounds that sport a glyceryl backbone linked by ester bonds with a polar phosphate head and two chains of fatty acids. The characteristics of these two moieties, along with the cholesterol percentage content, contribute to the properties of the final liposome preparation, such as permeability, phase transitions, hydrophilic–lipophilic balance (HLB), dimensions, and affinity to a variety of substrates.

Liposomes are classified by size and number of bilayers (Figure 2); the most used drug delivery systems are unilamellar nanoliposomes in the 50–200 nm range, depending on the targeted tissue [85].

Several preparation methods exist for these vehicles: the thin layer hydration—extrusion method and the Mozafari method [86,87]. Liposomes can encapsulate various active molecules since they can be highly optimized in size. They can naturally contain lipophilic and hydrophilic compounds as well, and the great range of preparation methods can account for the weaknesses of the drugs, e.g., thermolability, sterical instability, and susceptibility to solvents and pH.

Even if simple liposomes had great success as phospholipid carriers, the need for more precise targeting, escape mechanisms from macrophages, longer half-life, and other crucial factors made scientists develop functionalized drug delivery systems that could satisfy these necessities.

#### 1.2.2. Functionalized Liposomes

PEGylation, the coating of liposomes with a thin layer of polyethylene glycol or the insertion of long PEG chains in the nano lipid structure, is a wide used technique to make “stealth” nanoparticles that avoid human immune surveillance systems retaining the original biocompatibility: bringing in the macrophage example, the thin layer of PEG negates protein adsorption, thus inhibiting opsonization, whereas long PEG chains heavily affect phagocytosis thanks to sterical and structural incompatibilities with the pseudopods [88]. Both PEGylation methods, mostly when used in synergy, considerably improve liposome half-life, but they are not exempt from downsides: drug unloading can be more challenging when PEG is included in phospholipids, rendering targeting less efficient and side effects more frequent, and the human body can also produce anti-PEG IgM when blood comes in contact with the polymer for too long; most recent studies found that these inconveniences can be mitigated by the regulating PEG shedding rate [89].

Fusogenic liposomes, as their name suggests, boast a peculiar fusion mechanism with cellular membranes triggered by microenvironmental pH [90]. The specific phospholipid composition (that comprises a mixture of charged lipids necessary for the additional phase transition, the inverted hexagonal phase, responsible for the fusion) ensures that they display optimal stability in physiological conditions, but when they encounter an element whose extracellular pH is acidic—e.g., cells suffering from tumoral diseases or bacterial infections and the outer membrane of Gram-negative bacteria [91,92]—a conformational change in lipids disposition make them able to merge with biomembranes and to release their content selectively; however, this type of targeting is not absolute and the surface charge renders them more susceptible to faster excretion.

Liposomes can be modified with different types of molecules to achieve better targeting properties or to monitor their bio-distribution. Passive targeting can be obtained by conjugating molecules that naturally accumulate in specific sites, like biotin for breast cancer cells and folate for FR-rich tumoral regions [93,94]. Active targeting is mostly reached by decorating phospholipids with monoclonal antibodies and viral glycoproteins, which are engineered to reach very specific targets [95,96]. Conjugation and decoration come with the major downside of an improved immunogenic response from the host system and thus faster degradation/resistance reactions, which can be partially surpassed by combination with PEGylation (Figure 3).

#### 1.2.3. Exosomes

Extracellular vesicles (EVs) are subcellular structures delimited by a lipid bilayer and shed by cells into their surrounding environment. These vesicles are involved in cell-to-cell communication and play important roles in various physiological and pathological processes. They are further categorized in the following three main types of extracellular vesicles:Exosomes are the smallest type of EVs, typically ranging in size from 30 to 200 nanometers. They originate from the endocytic pathway within the cell. Exosomes contain a diverse array of biomolecules, including proteins, nucleic acids (such as RNA and DNA), and lipids.Microvesicles are larger than exosomes, typically ranging from 100 to 1000 nanometers in size. They are formed by the outward budding of the cell’s plasma membrane and contain a similar assortment of biomolecules, serving as carriers for intercellular communication.Apoptotic bodies are the largest of the extracellular vesicles, typically exceeding 1000 nanometers in size. They are produced during the process of cell apoptosis (programmed cell death) and contain cellular debris and organelles from the dying cell.

These three subtypes vary in size, function, biological origin, and other attributes.

Exosomes, which are the smaller extracellular vesicles (EVs), are accumulated into multi-vesicular bodies (MVBs) before secretion. Exosome are shed by a wide array of cell types, including dendritic cells (DCs), lymphocytes, epithelial cells, endothelial cells, neurons, etc. They are detectable in an extensive range of bodily fluids, including blood, urine, saliva, amniotic fluid, breast milk, hydrothoracic fluid, ascitic fluid, and the culture medium of most cell types [97]. Various factors can induce the release of exosomes: microbial agents, external stimuli, and various stressors can elicit these responses [98]. Exosomes encompass sugars, lipids, proteins, nucleic acids, and bioactive substances within the extracellular matrix. Initially, their role was believed to be the removal of metabolic waste [99].

The composition of exosomes differs depending on their source cell (Figure 1). So far, research has identified nearly 350,000 proteins, 40,000 nucleic acids, and 600 lipids in various exosomes [100]. This extensive flexibility offers numerous possibilities for practical clinical diagnosis and treatment scenarios involving exosomes.

Exosomes contain a wide range of proteins, including transmembrane proteins, lipid-anchored membrane proteins, peripherally adsorbed membrane proteins, and soluble proteins within the exosome lumen [101]. The most commonly found proteins in exosomes include membrane transporters and fusion proteins (e.g., GTPases, annexins, and flotillin), heat shock proteins (e.g., HSC70), tetraspanins (e.g., CD9, CD63, and CD81, which are typically considered exosome markers), proteins involved in multi-vesicular body (MVB) biogenesis (e.g., Alix and TSG101), and lipid-related proteins and phospholipases. Some proteins are recognized as specific markers of exosomes, with CD63 and CD81 tetraspanins being the most commonly used ones. Noteworthy, exosomes are also rich in lipids, primarily cholesterol, sphingolipids, phospholipids, and bisphosphates [102].

Exosomes are flexible enough to carry a variety of nucleic acids such as RNAs (mRNA, microRNA, and other non-coding RNAs) and DNAs (mitochondrial DNA, double-stranded DNA, single-stranded DNA, and viral DNA), suggesting that exosomes could act as carriers of genetic information [103]. Despite many of the RNAs in exosomes being degraded fragments with lengths of less than 200 nucleotides, some full-length RNAs may be present and delivered to recipient cells through endocytosis, potentially influencing protein production in those cells. In this context, exosomal miRNAs are also associated with specific diseases [104].

Beyond their cargo, exosomes exhibit an intricate array of biological agents on their membranes. Adhesion molecules, signaling molecules, immunomodulatory factors, receptors, antibodies, lipids, proteins, transporters, and channels collectively contribute to the complexity of these vesicles, facilitating their interaction with the target cells.

Recent attention has focused on exosomes as an auspicious drug delivery system. Their inherent biocompatibility, efficient delivery mechanisms, and minimal immunogenicity have elevated their standing. Extensive research has revealed the role of exosomes in mediating intercellular communication and participating in various physiological and pathological processes in the body. Their functions span a wide spectrum, encompassing immune responses, antigen presentation, cell migration, differentiation, angiogenesis, inflammation induction, apoptosis, atherosclerosis, tumor development, invasion, metastasis, and drug resistance. The capacity of exosomes to transport bioactive substances holds great potential for deciphering an enigma of diseases such as cancer, neurological disorders, cardiovascular conditions, and metabolic disorders, as well as for disease diagnosis using biomarkers [98,105].

Therefore, exosomes as a drug delivery system stand out for their several advantages over existing synthetic delivery systems like liposomes. Their reduced likelihood of provoking immune reactions and toxicity, coupled with their inherent precision in targeting specific cells, renders them an attractive choice. In particular, small RNA therapeutics, anti-inflammatory agents, and anticancer drugs are among the drugs that could particularly benefit from delivery via exosomes. Researchers have explored two approaches to loading exosomes with therapeutic small RNA molecules: post-loading after EV isolation (known as the exogenous method) and pre-loading during EV formation (known as the endogenous method). However, the effectiveness of these methods has yet to be fully demonstrated [106].

## 2. Innovative Phospholipid Carriers versus Antimicrobial Resistance

IPCs have found profuse employment in treating microbial infections as adjuvants and carriers, especially in therapeutic and theragnostic applications against resistant and multi-resistant bacteria. Starting from “simple” liposomes, some formulations were so successful and incisive that they even hit the market, revolutionizing the battle against certain illnesses.

Liposomal antibiotic preparations can be traced back to the 1990s when workgroups like Lagacé et al. tried to deal with complicate bacterial infections like the one sustained by Pseudomonas Aeruginosa. This notoriously difficult-to-deal-with microbe also tends to produce a potent biofilm and to internalize into the host cells: they found that the employment of a fairly plain distearoylphosphatidylcholine/1,2-dimyristoylphosphatidylglycerol 10:1 liposome dramatically enhanced the sensitivity of resistant P. Aeruginosa to ticarcillin and tobramycin [107].

In line with these results, other groups have investigated the potential amelioration of treatments against potent bacteria with the encapsulation of traditional, not-more-efficacious antibiotics in liposomal formulations, obtaining in this way promising results [108,109,110]: among them, a case study was represented by two liposomal ciprofloxacin products, Lipoquin and Pulmaquin, for inhalation use, which permits ciprofloxacin to be utilized again as a first-line choice when the patient is facing complicated lung infections, even in cystic fibrosis-based (CF) scenarios [111,112]; sadly, both formulations were discontinued in 2022 since decisive data on their functionality were not achieved after an ORBIT phase 3 study. A different, brilliant destiny was instead achieved by liposomal formulations of amikacin (Arikayce), which have been tested with remarkable outcomes in CF, particularly in non-CF severe lung illnesses alone, and in combination with colistin [81,113].

There are even particular cases in which non-antibiotic molecules, both alone or co-encapsulated with antibiotics, proved to be useful to treat collateral symptoms, to adjuvant in the circumvention of common resistance mechanisms such as MDR pumps, to obtain clear imaging and theragnostic effects, to control ocular microflora after cataract surgery, and to preemptively stimulate innate immunity for major protection from bacterial infections, hinting at the diverse out-of-the-box possibilities offered by liposomal preparations [114,115,116,117,118].

Since most bacteria present a negatively charged outer layer, cationic and fusogenic liposomes proved to possess a certain selectivity and thus better delivery thanks to their net positive charge, favoring electrostatic interactions but also, especially in the case of fusogenic liposomes, triggering the fusion mechanism only in that determined microenvironment [119,120]. Similarly to their basic counterpart, but with the added benefits described above, these liposomal preparations enhanced and broadened the spectrum of action of classic antibiotics [121,122] and have found interesting applications in photodynamic therapy disinfection and photo-inactivation with aluminum chloride phthalocyanine and a porphyrinic compound, respectively [123,124]. An interesting work also showed how the fusogenic abilities of these carriers can be amped up by further decoration with cell-penetrating peptides like HIV-derived Tat surface protein, hinting at the strong potential of liposomal functionalization in therapy [125].

The most remarkable goals in antimicrobial (and not) applications of nanotherapy were indeed reached by functionalized liposomes, especially antibody-decorated ones, thanks to their precise targeting and the possibility of combining this technology with other techniques such as PEGylation, which proved to be extremely useful on its own due to the masking and stabilizing properties exerted on liposomes [126]. Surface-engineered liposomes can use a variety of ligands to properly direct therapeutics to sites of interest, whether the target is bacteria themselves or tissues affected by them while adopting synergistic mechanisms like cell penetration, mucoadhesion, tetraether-lipid-based stabilization, hetero-multivalent targeting, and the previously cited PEGylation [127,128,129,130].

Antibody-conjugated liposomes (ACL) have been thoroughly tested in a variety of applications, ranging from imaging-guided theragnostic activity [131] and maintenance of bacterial homeostasis [132] to the targeted release of antibiotics and, more generally, antimicrobial molecules. Natural extracts with known antimicrobial properties, like clove essential oil, have been successfully enclosed in ACL and showed a more precise release but even prolonged activity and bacteria-concentration-dependent action [133]. A detailed study by Krivic et al. demonstrated how antibody-conjugated hybrid erythrocyte liposomes encapsulating polymyxin B were capable of maintaining the unaltered activity of this antibiotic; drastically reducing common side effects (in vitro model) like hemolysis and nephrotoxicity; considerably improving drug retention and half-life; and, finally, obtaining selectivity against certain bacterial strains, namely *E. coli* and *P. aeruginosa* [134]. ACL can even deliver novel antisense oligonucleotides (composed of nucleic acid mimics with antimicrobial properties, paving the way to additional therapeutic options [135].

Virosomes, while promising and innovative, are a rather new technology that has not been extensively tested in the antimicrobial field yet [136]. On the other hand, exosomes, also recently employed as a drug delivery system, have already obtained landmark results against resistant bacteria.

The higher biocompatibility and membrane complexity of exosomes and exosome-like vesicles make sure that these carriers achieve a better intracellular uptake, evading lysosomal degradation in the cytoplasm and, in some cases, inducing macropinocytosis, thanks also to avant-garde synergism with other techniques such as decoration with cell-penetrating peptides [137]. Loadings of exosomes with natural toxins, like bee venom and mycobacterial antigens, have proven efficacious in eradicating mortal *E. coli* K99 infections in calves and provoking immunization with antigen-specific IFN-gamma and lymphocytic response against *Mycobacterium tuberculosis*, respectively [138,139]. While certain exosomes alone proved to exert an immunomodulatory and anti-inflammatory effect in cells during bacterial infections, being then a great aid in the co-administration with regular antibiotics to reduce side effects and pathological inflammatory [140], the most interesting applications are indeed in the ameliorated suppression of MRSA-sustained infections by loading antibiotics, namely linezolid and vancomycin in conjunction with lysostaphin [141,142].

## 3. Discussion

Antimicrobial resistance is undoubtedly one of the major health threats that should be overcome in the field of infectious diseases. Since its etiologies are so broad and diverse in terms of implicated molecular mechanisms and since bacteria can rapidly exchange this kind of information with each other, even among different species, AMR represents a worldwide problem that should be eradicated to avoid the insurgence of MDR-bacteria-driven illnesses, the latter of which proved to be highly deadly and resilient to complete eradication [143].

Apart from the natural occurrence of the spontaneous selection of resistant bacterial strains, the augmented appearance of MDR bacteria in recent decades has been amped up principally by a steady misusage and over-usage of antibiotics, both in community and nosocomial settings: the longer, unnecessary exposure to the drug and/or the employment of inefficacious dosages of pharmaceutics considerably speed up the bacterial selection process, favoring the flourishment of multi-resistant species; furthermore, the constant presence of antibiotics in food and in the environment due to livestock-related malpractice and mishandling of industrial waste, respectively, led to additional reinforcement of dangerous selection and exchange processes that can easily make a great portion of antibiotics virtually useless [144].

Even if nowadays the cohorts principally subjected to strong and often deadly MDR-bacteria-led infections are somewhat small, mostly relegated to hospitals and represented by elder and immunodeficient people, an increasing number of victims between healthy, immunological-sturdy patients is showing up in recent times—an alarming signal of the marked virulence that these pathogens possess and could increasingly exert in the near future if not contained appropriately [145].

The principal exponents of MDR bacteria are indeed methicillin-resistant Staphylococcus aureus (MRSA), vancomycin-resistant *Staphylococcus aureus* (VRSA), Vancomycin-resistant *Enterococcus* (VRE), *Mycobacterium tuberculosis* MDR, extended-spectrum β-lactamase-producing *Enterobacterales* (ESBL), *Klebsiella pneumoniae* carbapenemase (KPC), and New Delhi Metallo-beta-lactamase-producing *Pseudomonas aeruginosa* (NDM) [146]. Through the development of resistance mechanisms that ensure bacterial survival against various concentrations of antimicrobial agents and other bacteria, genetic changes such as horizontal gene transfer by mobile genetic elements are to blame for the higher incidence of susceptibility loss, favoring the growth, colonization, and progression of the infectious process [145].

Strategies for overcoming these persistent infections include the following: the employment of multidrug protocols and pharmaceutics that do not trigger resistance mechanisms, increased social awareness, and correct use of conventional antibiotics, combined with vehicles or co-drugs that surpass specific resistance mechanisms [147]. Among these, IPCs, other than improving the overall pharmacokinetics of most antibiotics, both in terms of bettered metabolic profile and target selectivity, help in negating resistance mechanisms related to biofilm formation, outer membrane/plasmatic membrane modifications, and bacterial inclusion into host cells (i.e., *P. Aeruginosa* and *Mycobacterium tuberculosis*). This trend is proven beyond the existence of commercial or under clinical evaluation products (e.g., Arikayce, Lipoquin) and establishes the role of IPCs as a valid instrument against AMR [148].

## 4. Conclusions and Future Perspectives

The objective of this review is to underline the important synergy between non-/conventional antibiotics and IPCs in order to fight chemo-resistance phenomena properly, to achieve better targeting, and to reduce a plethora of annoying side effects; even if this research field is thriving, with some notable examples of pharmaceutical formulations which successfully hit the market, the necessity of carrying on more studies and experimentations on IPCs specifically tailored to carry antibiotics and other essential antimicrobial molecules is clearly inferable from the scientific evidence collected in this work. However, this alone obviously cannot be the solution to a complex problem such as AMR, which indeed needs to be approached in a multi-strategical and multidisciplinary way: starting from the conventional therapies, multi-therapy protocols should be properly adopted in conjunction with precise antibiograms, pondered dosages of antibiotics, and molecular diversification to reduce, at minimum, the risk of resistance derived by nosocomial and community misusage. The employment of antimicrobial peptides and other molecules (i.e., oligonucleotides, metals, etc.) that naturally do not trigger chemo-resistance mechanisms can definitely help reduce the burden on conventional antibiotics; other carriers like solid lipid nanoparticles (SLN) and nanostructured lipid carriers (NLC), and innovative biocarriers such as bacteriophages can offer additional solutions to formulation, chemical stability, and delivery problems; last but not least, the strong need for more and more social awareness in communities, hospitals, and industries regarding correct handling of antibiotics and the strong impact that multidrug-resistant superbugs (MSRBs) are already exerting on our health and environment should be communicated at local, national, and international levels.

To sum up, this review wants to be an additional voice in the already loud, worried chorus about the rampage of microbial resistance and its dangerous outcomes: if common action is not taken properly starting from the present, the world will experience pandemics much worse than the recent COVID-19 outbreak, as claimed by the World Health Organization (WHO), which predicts 5.2 million due to AMR deaths in the Western Pacific alone by 2030 [149]. In this scenario, IPCs are a well-known, precious weapon against AMR that should be implemented more, given the promising results it produced over time.

## Figures and Tables

**Figure 1 ijms-24-15934-f001:**
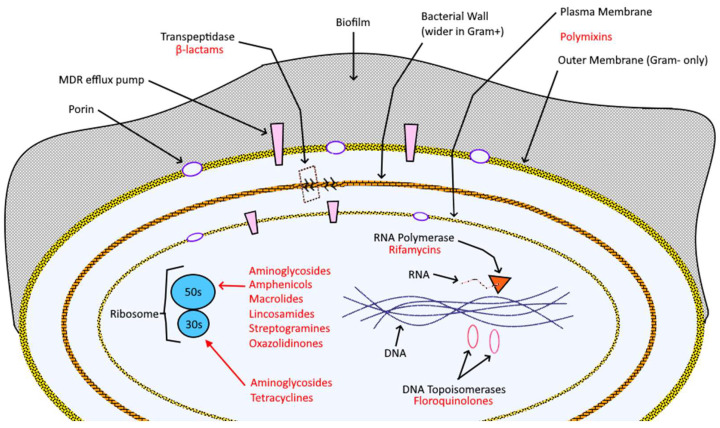
Bacterial cell composition, key structures (in black), and targeting antibiotics (red).

**Figure 2 ijms-24-15934-f002:**
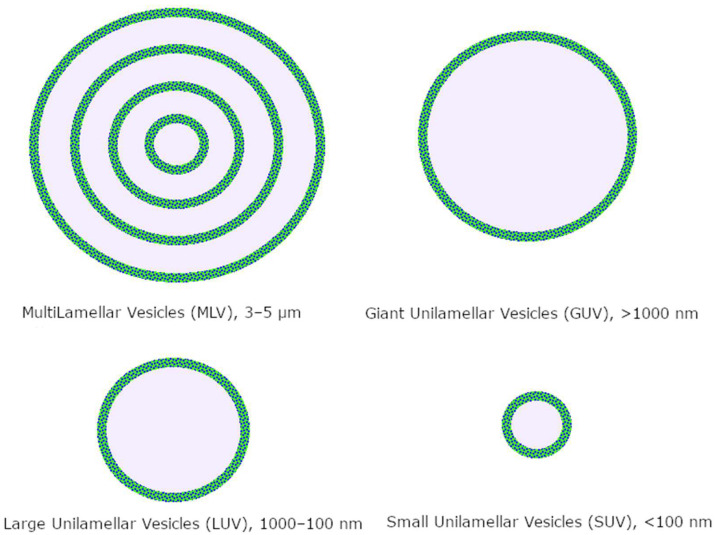
Classification of liposomes.

**Figure 3 ijms-24-15934-f003:**
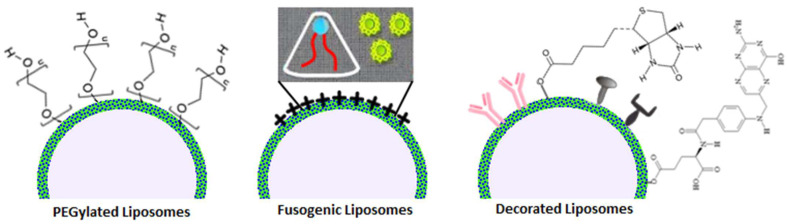
Visualization of different types of functionalized liposomes.

**Table 1 ijms-24-15934-t001:** Specific mechanism of action (M.O.A.) and chemo-resistance mechanisms (C.M.) per class of antibiotic, with references.

Articles	C.M.	M.O.A.	Antibiotic
[12,13]	β-lactamases (minor for Cephalosporins)	Inhibition of DD-TP and PBPs	Penicillins and Cephalosporins
[14,15,16]	Carbapenemases, target site mutation and efflux pumps	Inhibition of DD-TP and PBPs	Carbapenems, Monobactams and Penems
[17,18]	Target site mutation (methylation/de-methylation), membrane non-permeability, enzymatic degradation	Bond to 30S ribosomal subunit (sterical clash on A-site)	Tetracyclines
[19,20]	Target site mutation (methylation/de-methylation), membrane non-permeability, efflux pumps, enzymatic degradation	Bond to 30S–50S ribosomal subunits (link to h44–H69 sites, prevent translocation)	Aminoglycosides
[21,22]	Target site mutation (methylation/de-methylation), membrane non-permeability, enzymatic degradation	Bond to 50S ribosomal subunit (prevent peptide-bond formation on Cam1 (eu) and Cam2 (archea))	Amphenicols
[23,24,25]	Translation of specific “bottle brush” oligopeptides, bypass synthesis, target site mutation (methylation/de-methylation), membrane non-permeability, enzymatic degradation	Bond to 50S ribosomal subunit (block ribosomal exit tunnel)	Macrolides
[26,27,28]	Target site mutation (methylation/de-methylation), membrane non-permeability, enzymatic degradation	Bond to 50S ribosomal subunit (sterical impedance on A-site at peptidyl-transferase center)	Lincosamides
[29,30,31]	Translation of specific “bottle brush” oligopeptides, bypass synthesis, target site mutation (methylation/de-methylation), membrane non-permeability, enzymatic degradation	Bond to 50S ribosomal subunit (block ribosomal exit tunnel on specific, synergistic sites)	Streptogramins
[32,33,34]	Target site mutation (methylation/de-methylation), membrane non-permeability, enzymatic degradation, PoxtA- and OptrA-mediated resistance	Bond to 50S ribosomal subunit (bind to A-site, acting as initiation inhibitor)	Oxazolidinones
[35,36]	Mutations in rpoB encoding the β subunit of RNAP, inactivation by ADP-ribosylation, and other enzymatic degradations	Inhibition of DNA-dependent RNA synthesis (binding to prokaryotic RNA polymerases)	Rifamycins
[37,38,39]	Target site mutation (quinolone-resistance-determining regions), efflux pumps	Inhibition of bacterial DNA synthesis (blockage of two DNA bacterial topoisomerases)	Fluoroquinolones
[40,41]	Addition of cationic groups to LPS moieties, two-component signal transduction system enhancement, plasmid-encoded resistance determinant MCR-1	Disruption of bacterial membrane, inhibition of bacterial respiration	Polymixins

Bacteria exhibit intrinsic chemo-resistance (IC) to certain antibiotics due to their structural features (Gram+, Gram−, or mycobacteria) or their natural protein production (AmpC B-lactamase and multidrug resistant (MDR) efflux pumps), i.e., Gram− bacteria are traditionally resistant to hydrophilic drugs, namely macrolides, because of the presence of an outer membrane that sterically blocks said actives or pushes them out through MDR pumps [9]. Acquired chemo-resistance (AC) is either the strengthening of innate mechanisms, like B-lactamase-augmented production, or the development of new mechanisms, such as methicillin-resistant *Staphylococcus aureus* (MRSA). AC, differently from IC, can be passed over, even among different bacterial species, through horizontal gene transfer [42,43].

## Data Availability

No new data were created or analyzed in this study. Data sharing is not applicable to this article.

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
