# Peer review of "Innovative Phospholipid Carriers: A Viable Strategy to Counteract Antimicrobial Resistance"

_ijms, 2023, doi:10.3390/ijms242115934_

Round 1

Reviewer 1 Report

Comments and Suggestions for Authors

This manuscript broadly reviews the use of phospholipid carriers to counteract antimicrobial resistance. The topic is very interesting since innovative strategies to combat antimicrobial resistance is a health imperative today. Despite the undoubted interest of the subject, several important issues must be considered:

 - Regarding the format of the manuscript; Some parts are more related to an opinion article than a review article, as is the case of the value judgment on “financial cuts to health departments around the world”, which is not documented and is repeated in the introduction and second paragraph of page. 5.

-The use of figures in general and figures 3, 4 and 5 in particular is another important point. Figures in a text simplify complicated data and help you quickly understand what is written. In the case of the three figures mentioned, I do not find them useful in any way, they seem inappropriate to me. Regarding the rest of the figures, they are incomprehensible to the reader if they are not referenced in the text. Only allusion is made to figure 1 on page 10, in the last paragraph, and I am afraid that is not the figure that is intended to be referenced. There is no mention to the table in the text either.

- Regarding Table 1, the column where bacteria resistant to each type of antibiotic appears is unnecessary to me. Furthermore, it is not very clear in this column whether the authors intend to include all the species described for each resistance mechanism, in which case it is incomplete, or whether they want to include the most clinically relevant species, in which case, some of them should not be here. Bacterial names should include the full name of the genus if it is the first time they appear in the text.

-“Pneumococcus” is the common way to name the pathogen Streptococcus pneumoniae, as far as I know there is no genus called Pneumococcus. In Table 1, Pneumococcus spp appears several times. Does it refer to S. pneumoniae or Streptococcus spp?. In this Table 1, it appears sometimes together with Streptococcus pneumoniae and in others together with Streptococcus spp. please review it.

- On pages 15 and 16, second paragraph and fourth paragraph, respectively, they write Mycobacterium spp, and refer to Mycobacterium tuberculosis.

- Many of the acronyms are not defined in the text. For example: DHFR, HLB, DSPC, SLN, NLC, MSRB, AMR. Please, carefully review all of them.

-The title of Section 2: “IPCs vs AMR”, it cannot be understood as, the AMR acronym is not even defined previously. Titles should be clear and acronyms should be avoided for better understanding.  

- In this Section 2, a portion of the second paragraph is italicized.

-In the third paragraph of this Section 2, the authors mention two liposomal ciprofloxacin products, Lipoquin and Pulmaquin, the latter cited in reference 161. This reference is from 2016 and the article contemplated the “Design of a Phase 3 Program to Investigate Safety and Efficacy of Pulmaquin® in Non-Cystic Fibrosis Bronchiectasis (NCFBE) Patients Chronically Colonized with Pseudomonas Aeruginosa”. It would be of interest for the reader to know in which stage of development is this product nowadays.

-The text is over-referenced, some general knowledge ideas include up to 4 references in some cases. Authors must make the effort to choose the most representative, or the most current.

-On page 4, authors have forgotten to include the reference: “Acquired Chemo-resistance (AC) is either the strengthening of innate mechanisms, like B-lactamase augmented production [cite],”.

Comments on the Quality of English Language

Some parts of the manuscript require minor editing of the English.

Author Response

1 - Regarding the format of the manuscript; Some parts are more related to an opinion article than a review article, as is the case of the value judgment on “financial cuts to health departments around the world”, which is not documented and is repeated in the introduction and second paragraph of page. 5.

Authors’ answer: We thank the reviewer for pointing this out. We revised the entire manuscript, adding not only the examples provided by reviewer 1 but also those provided by reviewer 2. The new sentence reads as follows:.

-Page 1 from: “The action of antibiotics consists of stopping the proliferation (bacteriostatic) or kill- ing (bactericidal) the pathogen with the least affection affinity towards the cells of the host;” to:The antibiotics' therapeutic effect involves either inhibiting  proliferation (bacteriostatic) or killing  pathogen (bactericidal) with the lowest affinity for the host's cells

‘Page 2 the sentence: ….other than more social awareness and a crucial change of paradigm in the health department has been eliminated

-Page 5. From: “Moreover, most antibiotics are excreted unmodified from the animals or from industrial waste, and both byproducts are sadly poured into rivers, lakes and terrains, which become gyms for bacteria to train in. MDR bacteria can be consistently found in nature [80]–[83]. Add the hygiene- and protocol-related problems in hospitals (due to continuous financial cuts to health departments all over the world) to make up the perfect storm, with the rise of multi-resistant nosocomial infections that nowadays spread even out of hospitals to become serious community infections”. To: “Moreover, most antibiotics are excreted unmodified from the animals or from industrial waste, and both byproducts are sadly poured into rivers, lakes and terrains, which become gyms for bacteria to train in. MDR bacteria can be consistently found in nature [80]–[83]. Add the hygiene- and protocol-related problems in hospitals to make up the perfect storm, with the rise of multi-resistant nosocomial infections that nowadays spread even out of hospitals to become serious community infections.”

-Page 7 from: “Among them, two of the most used against microbes are solid lipid nanoparticles and liposomes; here it will be discussed the key features of the la?er in this paper the key features will be discussed, along with some of their technological evolutions, namely functionalized liposomes and exo- somes, under the label of Innovative Phospholipid Carriers (IPCs).

To:  Among them, the two types frequently used for the treatment of microbial infections are solid lipid nanoparticles and liposomes; this review will discuss the main features and some of their technological evolutions, namely functionalized liposomes and exosomes, under the label of Innovative Phospholipid Carriers (IPCs).

-Page 16. From: “To sum up, this review wants to be an additional voice in the already loud, worried chorus about the rampage of microbial resistance and its dangerous outcomes: if common action is not taken properly starting from the present, the world will experience pandemics much worse than recent COVID-19 outbreak; and while the most straightforward effort, as we all recently saw, should be a sound and constant financial sustain of the health department, we scientifically propose IPCs as a well-known, precious weapon against AMR that should be implemented more, given the promising results it produced over time.” To: “To sum up, this review wants to be an additional voice in the already loud, worried chorus about the rampage of microbial resistance and its dangerous outcomes: if common action is not taken properly starting from the present, the world will experience pandemics much worse than recent COVID-19 outbreak, as claimed by the World Health Organization (WHO), which predicts 5.2 million due to AMR deaths in the Western Pacific alone by 2030 (World Health Organization. (2023). Health and economic impacts of antimicrobial resistance in the Western Pacific Region, 2020-2030. In Health and economic impacts of antimicrobial resistance in the Western Pacific Region, 2020-2030.). In this scenario, IPCs are a well-known, precious weapon against AMR that should be implemented more, given the promising results it produced over time.”

Moreover, the abstract has been deeply re-written.

2 -The use of figures in general and figures 3, 4 and 5 in particular is another important point. Figures in a text simplify complicated data and help you quickly understand what is written. In the case of the three figures mentioned, I do not find them useful in any way, they seem inappropriate to me. Regarding the rest of the figures, they are incomprehensible to the reader if they are not referenced in the text. Only allusion is made to figure 1 on page 10, in the last paragraph, and I am afraid that is not the figure that is intended to be referenced. There is no mention to the table in the text either.

Authors’ answer: We thank the reviewer for pointing this out to us. This comment had also been raised by reviewer 2, thus we have ensured that all suggestions have been included in the manuscript. 

Figures 2, 3, 4, 5, 9 were eliminated from the text, and the numeration was re-arranged.

Moreover, the figures below were cross-referenced within the text:

Figure 1 pag. 2-10

Figure 2 pag. 8

Figure 3 pag. 9

Figure 4 pag. 10

Figure 5 pag. 12

Figure 6 pag. 14

Figure 7 pag. 14

Moreover, the following paragraph was moved up to allow the correct figure cross reference: “Since most bacteria present a negatively charged outer layer, cationic and fusogenic liposomes proved to possess a certain selectivity and thus better delivery thanks to their net positive charge, favoring electrostatic interactions but also, especially in the case of fusogenic liposomes, triggering the fusion mechanism only in that determined microenvironment (Figure 5). Similarly to their basic counterpart, but with the added benefits described above, these liposomal preparations enhanced and broadened the spectrum of action of classic antibiotics and have found interesting applications in photodynamic therapy disinfection and photo-inactivation with aluminum-chloride-phthalocyanine and a porphyrinic compound, respectively. An interesting work also showed how the fusogenic abilities of these carriers can be amped up by further decoration with cell-penetrating peptides like HIV-derived Tat surface protein, hinting at the strong potential of liposomal functionalization in therapy.”

3 - Regarding Table 1, the column where bacteria resistant to each type of antibiotic appears is unnecessary to me. Furthermore, it is not very clear in this column whether the authors intend to include all the species described for each resistance mechanism, in which case it is incomplete, or whether they want to include the most clinically relevant species, in which case, some of them should not be here. Bacterial names should include the full name of the genus if it is the first time they appear in the text.

Authors’ answer: Agree. This comment had also been raised by reviewer 2, thus we have revised Table 1

4  - “Pneumococcus” is the common way to name the pathogen Streptococcus pneumoniae, as far as I know there is no genus called Pneumococcus. In Table 1, Pneumococcus spp appears several times. Does it refer to S. pneumoniae or Streptococcus spp?. In this Table 1, it appears sometimes together with Streptococcus pneumoniae and in others together with Streptococcus spp. please review it.

Authors’ answer: Thank you for pointing this out. We agree with this comment. Therefore, according to the previous comment we have eliminated Resistant bacteria column from the Table

5 - On pages 15 and 16, second paragraph and fourth paragraph, respectively, they write Mycobacterium spp, and refer to Mycobacterium tuberculosis.

Authors’ answer: We agree with this and have incorporated your suggestion throughout the manuscript on pages 15 and 16, from Mycobacterium spp. to Mycobacterium tuberculosis

6 - Many of the acronyms are not defined in the text. For example: DHFR, HLB, DSPC, SLN, NLC, MSRB, AMR. Please, carefully review all of them.

Authors’ answer: We thank the reviewer for pointing this out to us. We agree with this and have defined all acronyms throughout the manuscript.

DHFR: DiHydroFolate Reductase pag. 6

HLB: Hydrophilic-Lipophilic Balance pag. 8

DSPC: Distearoylphosphatidylcholine pag. 11

SLN: Solid Lipid Nanoparticles pag. 16

NLC: Nanostructured Lipid Carriers pag. 16

MSRB: Multidrug Resistant SuperBug pag. 16

AMR: Anti-Microbial Resistance pag. 1

Cystic-Fibrosis-based (CF) page 11

DD-transpeptidase (DD-TPs) page 2

Penicillin-Binding Proteins, PBPs page 2

7 -The title of Section 2: “IPCs vs AMR”, it cannot be understood as, the AMR acronym is not even defined previously. Titles should be clear and acronyms should be avoided for better understanding.

Authors’ answer: We have made the change. The new sentence reads as follows: Innovative Phospholipid Carriers versus Anti-Microbial Resistance (section 2 pag. 11)

8 - In this Section 2, a portion of the second paragraph is italicized.

Authors’ answer: We agree with this and have incorporated your suggestion in section 2, second paragraph.

9 - In the third paragraph of this Section 2, the authors mention two liposomal ciprofloxacin products, Lipoquin and Pulmaquin, the latter cited in reference 161. This reference is from 2016 and the article contemplated the “Design of a Phase 3 Program to Investigate Safety and Efficacy of Pulmaquin® in Non-Cystic Fibrosis Bronchiectasis (NCFBE) Patients Chronically Colonized with Pseudomonas Aeruginosa”. It would be of interest for the reader to know in which stage of development is this product nowadays.

Authors’ answer: We agree with this comment. We have added the following sentence in the third paragraph of section 2 (page 11) “….sadly, both formulations were discontinued in 2022 since decisive data on their functionality was not achieved after an ORBIT Phase 3 study.” to emphasize this point.

10 - The text is over-referenced, some general knowledge ideas include up to 4 references in some cases. Authors must make the effort to choose the most representative, or the most current.

Authors’ answer: We thank the reviewer for pointing this out to us. We agree with this and have re-worked the entire bibliography throughout the manuscript.

11 -On page 4, authors have forgotten to include the reference: “Acquired Chemo-resistance (AC) is either the strengthening of innate mechanisms, like B-lactamase augmented production [cite],”.

Authors’ answer: Thank you for pointing this out. We agree with this comment. Therefore, we have added proper citation on page 4.

Comments on the Quality of English Language

Some parts of the manuscript require minor editing of the English.

Authors’ answer: all spelling and grammatical errors pointed out by the reviewers have been corrected, and an affiliation was added.  

Sincerely,

Stefano Russo

Reviewer 2 Report

Comments and Suggestions for Authors

The review by Nicolosi D et al has very valuable insights on the new drug delivery vesicles. Overall the content is good but it needs some minor edits/changes.

·      Figure 2, 3, 4, 5, 9 were not required for this paper.

·      The figures were not cross-referenced within the text to show which part of the text refers to it.

·      Table 1 could be reorganized in a  way to segregate the text better.

·      After each section of the IPCs, it would be better to include a closing statement on why each one has a promising role to play.

·      For the IPCs and AMR, introduce the abbreviation AMR a little earlier in the text, to make it easier for the reader to follow.

·      The title can be worded better since it sounds like a statement.

Comments on the Quality of English Language

The style of writing of some of the parts like the introduction and conclusion sounded casual. Also, there are some grammatical issues throughout the paper.

Just a few examples:

The action of antibiotics consists of stopping the proliferation (bacteriostatic) or kill- ing (bactericidal) the pathogen with the least affection affinity towards the cells of the host;”

Among them, two of the most used against microbes are solid lipid nanoparticles [123]–[126] and liposomes; here it will be discussed the key features of the la?er in this paper the key features will be discussed, along with some of their technological evolutions, namely functionalized liposomes and exo- somes, under the label of Innovative Phospholipid Carriers (IPCs).

Author Response

1 - Figure 2, 3, 4, 5, 9 were not required for this paper.

Authors’ answer: We thank the reviewer for pointing this out to us. This comment had also been raised by reviewer 1, thus we have ensured that all suggestions have been included in the manuscript. We have removed the selected Figures from the manuscript and re-arranged the numeration.

2 - The figures were not cross-referenced within the text to show which part of the text refers to it.

Authors’ answer: We thank the reviewer for pointing this out to us. This comment had also been raised by reviewer 1, thus we have ensured that all suggestions have been included in the manuscript. The figures below were cross-referenced within the tex text:

Figure 1 pag. 2-10

Figure 2 pag. 8

Figure 3 pag. 9

Figure 4 pag. 10

Figure 5 pag. 12

Figure 6 pag. 14

Figure 7 pag. 14

Moreover, the following paragraph was moved up to allow the correct figure cros reference: “Since most bacteria present a negatively charged outer layer, cationic and fusogenic liposomes proved to possess a certain selectivity and thus better delivery thanks to their net positive charge, favoring electrostatic interactions but also, especially in the case of fusogenic liposomes, triggering the fusion mechanism only in that determined microenvironment (Figure 5). Similarly to their basic counterpart, but with the added benefits described above, these liposomal preparations enhanced and broadened the spectrum of action of classic antibiotics and have found interesting applications in photodynamic therapy disinfection and photo-inactivation with aluminum-chloride-phthalocyanine and a porphyrinic compound, respectively. An interesting work also showed how the fusogenic abilities of these carriers can be amped up by further decoration with cell-penetrating peptides like HIV-derived Tat surface protein, hinting at the strong potential of liposomal functionalization in therapy .”

3 - Table 1 could be reorganized in a way to segregate the text better.

Authors’ answer: Agree. In line with reviewer 1 suggestion, we have revised Table 1 to emphasize this point.

4 - After each section of the IPCs, it would be better to include a closing statement on why each one has a promising role to play.

Authors’ answer:  We thank the reviewer for the comment. The second section serve as an explanatory chapter about the pharmacokinetic and biotechnological properties of the examined carriers; the point raised by the reviewer is already broadly covered in the third chapter.

5 - For the IPCs and AMR, introduce the abbreviation AMR a little earlier in the text, to make it easier for the reader to follow.

Authors’ answer:  We thank the reviewer for pointing this out. We have revised the suggested abbreviations a little earlier in the text:

pag. 1: Anti-Microbial Resistance (AMR)

pag.2 Innovative Phospholipid Carriers demonstrated (IPCs)

6 - The title can be worded better since it sounds like a statement.

Authors’ answer: Thank you for pointing this out. We agree with this comment. Therefore, we have changed the title from: Innovative Phospholipid Carriers and the Vision as a solid strategy to Counteract Anti-Microbial Resistance to: Innovative Phospholipid Carriers: A Viable Strategy to counteract Anti-Microbial Resistance

7 - Comments on the Quality of English Language

The style of writing of some of the parts like the introduction and conclusion sounded casual. Also, there are some grammatical issues throughout the paper.

Authors’ answer:  We thank the reviewer for pointing this out. We revised the entire manuscript, adding not only the examples provided by reviewer 2 but also those provided by reviewer 1. The new sentence reads as follows:.

-Page 1 from: “The action of antibiotics consists of stopping the proliferation (bacteriostatic) or kill- ing (bactericidal) the pathogen with the least affection affinity towards the cells of the host;” to:The antibiotics' therapeutic effect involves either inhibiting  proliferation (bacteriostatic) or killing  pathogen (bactericidal) with the lowest affinity for the host's cells

-Page 2 the sentence: ….other than more social awareness and a crucial change of paradigm in the health department has been eliminated.

-Page 5. From: “Moreover, most antibiotics are excreted unmodified from the animals or from industrial waste, and both byproducts are sadly poured into rivers, lakes and terrains, which become gyms for bacteria to train in. MDR bacteria can be consistently found in nature. Add the hygiene- and protocol-related problems in hospitals (due to continuous financial cuts to health departments all over the world) to make up the perfect storm, with the rise of multi-resistant nosocomial infections that nowadays spread even out of hospitals to become serious community infections”. To: “Moreover, most antibiotics are excreted unmodified from the animals or from industrial waste, and both byproducts are sadly poured into rivers, lakes and terrains, which become gyms for bacteria to train in. MDR bacteria can be consistently found in nature. Add the hygiene- and protocol-related problems in hospitals to make up the perfect storm, with the rise of multi-resistant nosocomial infections that nowadays spread even out of hospitals to become serious community infections.”

-Page 7 from: “Among them, two of the most used against microbes are solid lipid nanoparticles and liposomes; here it will be discussed the key features of the la?er in this paper the key features will be discussed, along with some of their technological evolutions, namely functionalized liposomes and exo- somes, under the label of Innovative Phospholipid Carriers (IPCs).

To:  Among them, the two types frequently used for the treatment of microbial infections are solid lipid nanoparticles and liposomes; this review will discuss the main features and some of their technological evolutions, namely functionalized liposomes and exosomes, under the label of Innovative Phospholipid Carriers (IPCs).

-Page 16. From: “To sum up, this review wants to be an additional voice in the already loud, worried chorus about the rampage of microbial resistance and its dangerous outcomes: if common action is not taken properly starting from the present, the world will experience pandemics much worse than recent COVID-19 outbreak; and while the most straightforward effort, as we all recently saw, should be a sound and constant financial sustain of the health department, we scientifically propose IPCs as a well-known, precious weapon against AMR that should be implemented more, given the promising results it produced over time.” To: “To sum up, this review wants to be an additional voice in the already loud, worried chorus about the rampage of microbial resistance and its dangerous outcomes: if common action is not taken properly starting from the present, the world will experience pandemics much worse than recent COVID-19 outbreak, as claimed by the World Health Organization (WHO), which predicts 5.2 million due to AMR deaths in the Western Pacific alone by 2030 (World Health Organization. (2023). Health and economic impacts of antimicrobial resistance in the Western Pacific Region, 2020-2030. In Health and economic impacts of antimicrobial resistance in the Western Pacific Region, 2020-2030.). In this scenario, IPCs are a well-known, precious weapon against AMR that should be implemented more, given the promising results it produced over time.”

In addition to the above comments, all spelling and grammatical errors pointed out by the reviewers have been corrected, the abstract was deeply re-worded, and an affiliation was added. 

Sincerely,

Stefano Russo

Round 2

Reviewer 1 Report

Comments and Suggestions for Authors

The manuscript has improved with the changes. There are minor considerations to take into account:

-On page 2, last paragraph, replace "Figure and Table 1" with "Figure 1 and Table 1".

-On page 5 section 1.2. The title should not include acronyms. This acronym has been previously defined.

-On page 6. The acronym Polyethylene Glycol should appear.

On pages 12 and 13, Mycobacterium tuberculosis should be replaced by M. tuberculosis since this is not the first time it appears.